# Psychosocial Correlates of Insomnia Symptoms Among Women and Men Receiving Buprenorphine Treatment for Opioid Use Disorder

**DOI:** 10.3390/neurosci6020048

**Published:** 2025-05-30

**Authors:** Sajanee Chithranjan, Michelle Eglovitch, Madison M. Marcus, Dace Svikis, Maha Alattar, Caitlin E. Martin

**Affiliations:** 1Department of Obstetrics and Gynecology, School of Medicine, Virginia Commonwealth University, Richmond, VA 23298, USA; 2Department of Psychology, College of Humanities and Sciences, Virginia Commonwealth University, Richmond, VA 23284, USA; 3Department of Neurology, School of Medicine, Virginia Commonwealth University, Richmond, VA 23298, USA; 4Institute for Drug and Alcohol Studies, Virginia Commonwealth University, Richmond, VA 23298, USA

**Keywords:** insomnia, sleep disturbance, buprenorphine, substance use disorders

## Abstract

Insomnia is common in individuals with opioid use disorder (OUD). Biopsychosocial factors are important in sleep health, yet this intersection has yet to be fully elucidated in people on buprenorphine for OUD. The objective is to report on patient-reported biopsychosocial factors among people with and without insomnia, specifically among women and men in outpatient OUD treatment. The parent study enrolled adults stabilized on buprenorphine from February 2022–September 2023. Scores of ≥11 on the Insomnia Severity Index (ISI) indicated clinically significant insomnia. Differences were detected by the presence of insomnia, stratified by men and women, using chi-squared and Fisher’s exact tests. Of the overall participants (N = 130), most (*n* = 77; 59.2%) met the criteria for clinically significant insomnia. Women with insomnia were more likely to report social stressors including discrimination for substance use (*p* = 0.040), food insecurity (*p* = 0.032), and transportation difficulties accessing healthcare (*p* = 0.043) than women without insomnia. Men with insomnia were more likely to report financial difficulties accessing healthcare (*p* = 0.023) than men without insomnia. These findings provide a unique perspective to consider in the development and implementation of sleep interventions for women and men receiving medication treatment for OUD.

## 1. Introduction

The overdose crisis is a multifaceted issue that continues to affect a significant portion of the population. In 2021, over 5 million people aged 12 years and older had an opioid use disorder (OUD) in the United States [1]. Further, opioid-involved overdose deaths increased from 49,860 in 2019 to 81,806 in 2022 [2]. Medications for opioid use disorder (MOUD), such as buprenorphine and methadone, are regarded as a first-line treatment for opioid use disorder, and are associated with significant reductions in overdose rates and better health outcomes [3]. Both methadone and buprenorphine are equally effective medications at achieving positive OUD treatment outcomes. In the United States, buprenorphine can be prescribed in an outpatient clinic setting, as opposed to methadone, which can only be dispensed by outpatient treatment programs abiding by substantial federal regulations. Thus, the expansion of buprenorphine access and utilization remains a critical public health strategy to address the ongoing overdose crisis, given the significant reduction in overdose risk that OUD treatments with medications (i.e., methadone, buprenorphine) render, compared to OUD treatments without medication [4].

Despite such advancements, the presence of sleep disturbances is a known barrier to MOUD initiation and maintenance [5]. A higher number of treatment attempts are linked to persistent sleep disturbances, implying that sleep disturbances may relate to recovery efforts [6]. In fact, patients with OUD have reported sleep disturbance as a main contributor to return to use and treatment discontinuation [6,7]. Recent data demonstrate that addressing sleep disturbances early in OUD treatment has been correlated with lower return-to-use rates (percentage of individuals who resume substance use after treatment at one, three, and six months post-treatment episode) [8]. There are multiple potential neurobiological mechanisms that underlie the relationship between OUD and sleep. Sleep disturbances in this population is associated with neurofunctional impairments, such as stress intolerance [9]. A study found that, for individuals taking opioids, sleep disturbances have a stronger impact on perceived stress compared to individuals using other substances [10]. Also, the orexin/hypocretin pathway may serve as a shared biological link between sleep disturbances and stress in the OUD population, as opioids can activate orexin neurons which additionally are known to impact sleep and stress outcomes [9].

Given the evidence indicating that poor sleep may negatively affect OUD treatment outcomes, it is important to identify factors that might impact sleep among people with OUD. Individuals with OUD experience unique psychosocial challenges, which may affect their sleep. Factors such as psychiatric comorbidity, pain, and substance use are known to negatively impact sleep outcomes in OUD patients; however, few studies have examined the association between sleep and psychosocial factors, such as social context, in this patient population [11]. The studies that do exist have produced mixed findings, suggesting that the intersection of psychosocial factors and sleep health in individuals with OUD has yet to be fully explored [11].

Utilizing sex-informed approaches to these investigations is especially important, as women and men face unique biopsychosocial factors that affect both their sleep quality and OUD recovery [12,13]. Understanding and addressing these biopsychosocial factors is important, as they can affect both the severity and persistence of sleep disturbances. Further, they may affect the patient’s ability to participate in and adhere to OUD treatment. Understanding how these factors may relate to sleep disturbances in people taking medication for opioid use disorder may ultimately inform the development of tailored, patient-centered OUD interventions.

The current study aims to explore comparisons of patient-reported psychosocial factors between individuals with and without insomnia among patients receiving buprenorphine in outpatient OUD treatment. A secondary objective is to examine psychosocial factors and insomnia separately among women and men on buprenorphine.

## 2. Methods

### 2.1. Participants and Study Design

This is a secondary analysis of data from a cross-sectional survey and medical record abstraction study, analyzing the relationship between insomnia symptoms and treatment of OUD in an outpatient addiction treatment facility. The clinic is associated with a large academic medical center in a southern, Medicaid-expanded state that works as a safety net program for low-income patients, with a majority identifying as being from a marginalized race or ethnicity. At this facility, medical providers prescribe buprenorphine for patients with OUD (methadone is not dispensed at this clinic), and mental health and recovery-oriented wraparound services are offered to all patients. Participants were recruited for the study through research assistants, who approached eligible patients in the clinic, verified participant eligibility, and invited them to participate. Informed consent was obtained from all subjects involved in the study. Comprehensive study methods have been previously published [14].

Sex refers to biological characteristics, and is typically assigned at birth, while gender is attributed to socially constructed roles and identities that may or may not align with their sex. All parent study participants identified as cisgender; therefore, the terms women and men are used in the present study to refer to cisgender women and men.

### 2.2. Measures

#### 2.2.1. Insomnia

Insomnia symptomatology was measured using the Insomnia Severity Index (ISI), which comprises seven items assessing insomnia symptomatology, rated on a 5-point Likert scale (ranging from 0, signifying no symptoms, to 4, signifying very severe symptoms). Scores were totaled and ranged from 0 to 28, with higher scores indicating higher levels of insomnia symptomatology. A threshold score of ≥11 was used to determine clinically significant insomnia [15].

#### 2.2.2. Psychosocial Factors and Demographics

Psychosocial factors were adapted from the PhenX Toolkit’s survey items [16]. In addition, the Socioecological Model of Sleep (SEM) was utilized to map variables onto three domains (individual, family, and sociocultural factors) [17]. Under individual factors, psychiatric comorbidities (depression, anxiety/panic disorder, PTSD, and bipolar disorder) and discrimination based on substance use, gender, and race were evaluated. Family factors such as socioeconomic status, health literacy, access to resources, neighborhood safety, and access to healthcare were examined. Demographic items included age, self-reported race, insurance status, education, and employment.

### 2.3. Data Analysis

Analyses were conducted with SPSS (version 28). We describe differences between participants with and without insomnia, stratified by men and women, using chi-squared and Fisher’s exact tests. For this exploratory study, no adjustments for multiple comparisons were performed.

## 3. Results

Participants (N = 130) were primarily women (*n* = 78; 60.0%) and identified as white (*n* = 72; 55.4%) (Table 1). Many participants were unemployed (*n* = 60; 46.2%) and enrolled in public insurance (*n* = 111; 85.4%). Over half of the participants (*n* = 77; 59.2%) met the criteria for clinically significant insomnia.

Women on buprenorphine with insomnia were more likely to have a PTSD diagnosis (*p* = 0.040), and report experiencing discrimination related to substance use disorder (SUD) (*p* = 0.040) than women without insomnia (Table 2). Additionally, women with insomnia were more likely to report food insecurity (*p* = 0.032) and greater difficulty accessing healthcare (*p* = 0.043) compared to women without insomnia. For men on buprenorphine, both with and without insomnia, similar patterns emerged across individual and family factors. However, men with insomnia were more likely to report forgoing medical care due to cost (*p* = 0.023) compared to men without insomnia.

## 4. Discussion

This study explored patient-reported psychosocial factors between people with and without insomnia on buprenorphine for OUD. Insomnia is a particularly pressing issue among people on MOUD, and a lack of sleep may relate to an increased risk of substance use recurrence and reduced quality of life. Applying the Social Ecological Model (SEM) provides a framework to understand how individual, community, and environmental factors may relate to health outcomes for OUD. The model focuses on the various levels, from individual to structural factors, that can influence health outcomes. In this sample of people receiving outpatient OUD treatment with buprenorphine, women and men varied slightly in their patterns of reported differences in psychosocial factors by their insomnia status.

In our sample, we found that PTSD was more commonly reported by women with insomnia compared to women without insomnia, whereas the same pattern was not observed in men. Studies have shown that there is much overlap between PTSD and sleep disturbances in OUD populations, with women experiencing this connection more acutely due to higher PTSD prevalence and psychosocial factors such as trauma exposure [7,18]. Additionally, both women and men reported high levels of experience with substance use and race-related stigma. Discrimination and stigma are known chronic stressors that can lead to greater sleep disturbances, and exacerbate mental health diagnoses [19]. For individuals with OUD, discrimination may be a barrier to treatment due to fear of judgment, and may prevent individuals from seeking or continuing their OUD treatment [20,21]. This is particularly true for marginalized communities among patients on buprenorphine, who experience compounded stigma due to intersecting identities such as gender, race, and socioeconomic status [22]. Experiencing discrimination may lead to a cycle where people on buprenorphine for their OUD avoid healthcare services which may exacerbate certain conditions, including insomnia, leaving them undertreated and ultimately further worsening their overall physical and mental health.

When looking at family factors, women on buprenorphine with insomnia were more likely to report food and transportation barriers, and men on buprenorphine with insomnia were more likely to endorse being unable to access care due to cost. These findings are consistent with the existing literature, which demonstrates that women are more likely than men to experience food and transportation insecurity, which stems from an interplay of economic, social, and cultural factors [23,24]. A major predictor of food insecurity is low socioeconomic status, along with factors related to caregiving, household management, and difficulties accessing education and employment; these gendered factors can limit the overall financial ability of women with OUD [25]. Thus, further research could explore how men and women on MOUD with comorbid health conditions are uniquely impacted by social determinants in health outcome trajectories, as compared to the general population.

It is important to note that psychosocial challenges were endorsed across all groups of men and women, including socioeconomic, financial, and healthcare challenges. A recent study has analyzed the role psychosocial factors can have on health outcomes in patients on MOUD with HIV, with positive social determinants of health including housing stability, no recent criminal justice involvement, a high-school level education or greater, and financial security all being associated with a decrease in baseline opioid use and improved opioid use outcomes [26]. Thus, this study supports the notion that future interventions for people on MOUD targeting sleep disturbances or other comorbid conditions should consider how to strengthen upstream factors and broader social, economic, and environmental contexts that could positively promote health outcomes.

The findings of this study align with existing literature, suggesting that psychosocial factors may be associated with poor sleep in patients on buprenorphine. Previous studies have highlighted how these factors contribute to poor sleep quality [11]. This study provides a unique perspective by reporting on differences in stressors experienced by patients with OUD and sleep disturbances, separately for women and men. The findings of this study could inform further research and the subsequent development of tailored sex-specific approaches to address the unique biopsychosocial stressors that people with OUD experience. A limitation of this study includes a reliance on cross-sectional self-reported data, which limits the ability to explore how psychosocial factors influence the persistence of sleep disturbances in patients with OUD. Additionally, self-reported data can be impacted by recall and misunderstanding of the questions. Finally, the findings are limited by this exploratory study’s small sample size and lack of corrections for multiple comparisons. Thus, significant findings should be interpreted with caution.

## 5. Conclusions

Men and women with insomnia also receiving outpatient OUD treatment are more likely to report psychosocial challenges than patients without insomnia. Further understanding of the role that social determinants of health and other biopsychosocial factors may play in sleep disturbances for women and men on MOUD could provide a patient-centered perspective to inform the creation of integrative sleep interventions faced by individuals with OUD. Future work should continue to explore such factors and how these factors can be addressed in precision medicine interventions to meet the unique biopsychosocial needs of men and women with OUD.

## Figures and Tables

**Table 1 neurosci-06-00048-t001:** Demographics of the study sample.

	Women N = 78		Men N = 52	
	With Insomnia (*n* = 50)	Without Insomnia (*n* = 28)	With Insomnia (*n* = 27)	Without Insomnia (*n* = 25)
Age (mean years ± STD)	37.8 ± 9.2	36.5 ± 9.3	43.4 ± 9.5	44.5 ± 8.9
Race				
White	32 (64.0%)	16 (57.1%)	13 (48.1%)	11 (44.0%)
Black	14 (28.0%)	9 (32.1%)	10 (37.0%)	13 (52.0%)
Other	3 (6.0%)	3 (10.7%)	4 (14.8%)	1 (4.0%)
Insurance				
Public	44 (88.0%)	25 (89.3%)	22 (81.5%)	20 (80.0%)
Private	3 (6.0%)	3 (10.7%)	1 (3.7%)	4 (16.0%)
Other	2 (4.0%)	0 (0%)	4 (14.8%)	1 (4.0%)
Employment status				
Employed	15 (30.0%)	7 (25.0%)	14 (51.9%)	10 (40.0%)
Unemployed	24 (48.0%)	16 (57.1%)	11 (40.7%)	9 (36.0%)
Disabled	10 (20.0%)	5 (17.9%)	2 (7.4%)	6 (24.0%)
Education				
<High school	7 (14.0%)	8 (28.6%)	4 (14.8%)	9 (36.0%)
High school/GED equivalent	24 (48.0%)	12 (42.9%)	12 (44.4%)	10 (40.0%)
>High school	18 (36.0%)	8 (28.6%)	11 (40.7%)	6 (24.0%)

**Table 2 neurosci-06-00048-t002:** Psychosocial characteristics of the study sample.

	Study Sample Characteristics	Women N = 78			Men N = 52		
		With Insomnia (*n* = 50)	Without Insomnia (*n* = 28)	*p* Value	With Insomnia (*n* = 27)	Without Insomnia (*n* = 25)	*p* Value
Individual factors	**Comorbidities**						
	Depression	38 (76.0%)	20 (71.4%)	0.657	12 (44.4%)	14 (56.0%)	0.405
	Anxiety/panic	42 (84.0%)	19 (67.9%)	0.098	17 (63.0%)	10 (40.0%)	0.098
	PTSD	14 (28.0%)	2 (7.1%)	**0.040**	4 (14.8%)	3 (12.0%)	0.776
	Bipolar disorder	12 (24.0%)	2 (7.1%)	0.063	3 (11.1%)	1 (4.0%)	0.336
	**Discrimination**						
	For subst. use	44 (88.0%)	18 (64.3%)	**0.040**	22 (81.5%)	17 (68.0%)	0.569
	For gender	27 (54.0%)	10 (35.7%)	0.577	7 (25.9%)	6 (24.0%)	0.839
	For race	34 (68.0%)	17 (60.7%)	0.601	19 (70.4%)	10 (40.0%)	0.666
Family factors	Socioeconomic status						
	*In the last 12 months, did you ever eat less than you felt you should because there wasn’t enough money for food?*	24 (48.0%)	6 (21.4%)	**0.032**	15 (55.6%)	10 (40.0%)	0.281
	*In the last 12 months, have you needed to see a doctor, but could not because of cost?*	10 (20.0%)	1 (3.6%)	0.075	10 (37.0%)	2 (8.0%)	**0.023**
	Health literacy *Do you ever need help reading hospital materials?*	7 (14.6%)	1 (3.6%)	0.238	5 (18.5%)	3 (12.0%)	0.227
	Resources (home, bedroom) *Are you worried that in the next 2 months, you may not have stable housing?*	20 (40.0%)	6 (21.4%)	0.159	11 (40.7%)	7 (28.0%)	0.190
	Residential Safety and Security *Are you afraid you might be hurt in your apartment building or house?*	2 (4.0%)	2 (7.4%)	0.424	2 (7.4%)	0 (0%)	0.382
	Access to healthcare *In the last 12 months, have you ever had to go without healthcare because you didn’t have a way to get there?*	16 (32.0%)	2 (7.4%)	**0.043**	3 (11.1%)	7 (28.0%)	0.096
Neighborhood and broader socio-cultural factors	*I feel safe walking in my neighborhood, day or night*	38 (76.0%)	22 (81.5%)	0.979	24 (88.9%)	21 (84.0%)	0.692

## Data Availability

The data presented in this study are available on request from the corresponding author due to the sensitive nature of data from people in active treatment for substance use disorder.

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
