# Peer review of "Psychosocial Correlates of Insomnia Symptoms Among Women and Men Receiving Buprenorphine Treatment for Opioid Use Disorder"

_neurosci, 2025, doi:10.3390/neurosci6020048_

Round 1
Reviewer 1 Report
Comments and Suggestions for Authors
This brief report by Chithranjan et al discussed the correlation between insomnia and receiving treatment for opioid use disorder. It is of high novelty and significance to the field, with minor changes needed before proceeding to publication.
There needs to be more discussion regarding the gender differences. For instance, the authors found that women with PTSD are significantly more susceptible to insomnia than men; men ONLY showed significant correlation with insomnia when cannot seek medical care due to the cost. In general, women are more susceptible to insomnia compared to men. More elaborated discussion is needed to provide insights of the cause.
When listing different discriminations, 'neighborhood cohesion' and safety is only one subcategory under 'family factors' while 'neighborhood and broader socio-cultural factors' is its own category. The overlapping of these two factors ('neighborhood cohesion' and 'neighborhood and broader socio-cultural factors') are too much that should not be considered as two independent factor.
This manuscript lack mechanistic explanations to correlate chronic stressors, OUD treatment and insomnia. More detailed molecular biology and neuroscience content need to be provided.
Author Response
There needs to be more discussion regarding the gender differences. For instance, the authors found that women with PTSD are significantly more susceptible to insomnia than men; men ONLY showed significant correlation with insomnia when cannot seek medical care due to the cost. In general, women are more susceptible to insomnia compared to men. More elaborated discussion is needed to provide insights of the cause.
- We appreciate this comment from the reviewer. The following has been added to the discussion on page 5: In our sample, we found that PTSD was more commonly reported by women with insomnia compared to women without insomnia, whereas the same pattern was not observed in men. Studies have shown that there is much overlap between PTSD and sleep disturbances in OUD populations, with women experiencing this connection more acutely due to higher PTSD prevalence and psychosocial factors such as trauma exposure [18,19].
- In addition, the following is on page 6 regarding gender differences: When looking at family factors, women on buprenorphine with insomnia were more likely to report food and transportation barriers, and men on buprenorphine with insomnia were more likely to endorse being unable to access care due to cost. These findings are consistent with existing literature that demonstrates that women are more likely than men to experience food and transportation insecurity, which stems from an interplay of economic, social, and cultural factors [24,25]. A major predictor of food insecurity is low socioeconomic status, along with factors related to caregiving, household management, and difficulties accessing education and employment; these gendered factors can limit the overall financial ability of women with OUD [26].
When listing different discriminations, 'neighborhood cohesion' and safety is only one subcategory under 'family factors' while 'neighborhood and broader socio-cultural factors' is its own category. The overlapping of these two factors ('neighborhood cohesion' and 'neighborhood and broader socio-cultural factors') are too much that should not be considered as two independent factor.
- We agree that this category is duplicative. The neighborhood cohesion factor has been renamed ‘residential safety and security’ in table 2 on page 4.
This manuscript lack mechanistic explanations to correlate chronic stressors, OUD treatment and insomnia. More detailed molecular biology and neuroscience content need to be provided.
- The following mechanistic explanations have been added on page 2: There are multiple potential neurobiological mechanisms that underlie the relationship between OUD and sleep. Sleep disturbances in this population is associated with neurofunctional impairments such as stress intolerance [9]. A study found that for individuals taking opioids, sleep disturbances have a stronger impact on perceived stress compared to individuals using other substances[10]. Also, the orexin/hypocretin pathway may serve as a shared biological link between sleep disturbances and stress in the OUD population, as opioids can activate orexin neurons which additionally are known to impact sleep and stress outcomes [9].
Reviewer 2 Report
Comments and Suggestions for Authors
This brief report addresses an important and relevant topic: the intersection of psychosocial factors, insomnia, and opioid use disorder (OUD) treatment with buprenorphine, specifically examining potential sex differences. The study utilizes a relevant framework (Socioecological Model of Sleep) and standard measures (ISI). The findings highlighting differences in correlates of insomnia between women (PTSD, SUD discrimination, food insecurity, transport barriers) and men (financial barriers to healthcare) are potentially valuable for informing tailored interventions. However, several issues related to reporting clarity, statistical considerations, and minor errors need attention to improve the manuscript.
- Page 1, Abstract: "February 2022-September 2023" - Clarify if this is the data collection period for the parent study or the period during which this secondary analysis occurred. Data collection period seems more relevant.
- The authors are required to justify why did they particularly chose Buprenorphine in this study although they mentioned that both Buprenorphine and methadone are the first line treatment for OUD in the introduction section. It is also preferred to give a brief introduction about Buprenorphine usage and the nature and side effects of the drug.
- There are multiple inconsistencies in the reported sample sizes throughout the manuscript, which significantly hinders clarity and reproducibility.
- Abstract: States N=127 overall participants.
- Results Text (Page 4): States "Participants (N=130) were primarily women (n=78; 60.0%)..." Calculation: 78/130 = 60%. Okay.
- Table 1 Header: States Women N = 77, Men N = 52. Total = 77 + 52 = 129.
- Table 1 Data: Sum of women with/without insomnia = 49 + 28 = 77. Sum of men with/without insomnia = 27 + 25 = 52. Total = 129.
- Table 2 Header: States Women N = 75, Men N = 52. Total = 75 + 52 = 127.
- Table 2 Data (Women): Sum of women with/without insomnia = 48 + 27 = 75. Matches header.
- Abstract & Results Text (Insomnia numbers): Abstract states n=75 (59.0%) met criteria for insomnia. Results text (page 4) states n=75 (59.0%). However, adding the numbers from Table 1 (which aligns with the total N=129) gives 49 women + 27 men = 76 participants with insomnia.
- Action Required: The authors must carefully reconcile these numbers. Please verify the final analytic sample size (is it 127, 129, or 130?) and ensure all reported numbers (total N, N for women, N for men, N with/without insomnia, Ns in tables) are consistent throughout the Abstract, Methods, Results text, and Tables. Explain any participant exclusions that might lead to differing Ns between analyses or tables, if applicable. The discrepancy between 75 and 76 participants meeting insomnia criteria also needs correction and consistent reporting.
- The authors appropriately state in the Methods (Section 2.3) that no adjustments for multiple comparisons were made due to the exploratory nature of the study. This limitation should be more strongly emphasized in the Discussion section. Acknowledge that the significant findings (p < .05) should be interpreted with caution due to the lack of correction and warrant replication in larger studies. Briefly mentioning the number of tests performed could also add context for the reader regarding the potential for chance findings.
- Page 1, Introduction, Para 4, Sentence 3: "...main contributor to return to use and treatment discontinuation [6] [5]." The previous sentence references factors linked to treatment attempts [5]. It seems possible the order should be [5, 6] here as well. Please double-check that the content cited aligns precisely with the reference numbers used.
- Also in Page 5, Discussion, Para 3, Sentence 3: "...interplay of economic, social, and cultural factors [19] [20]."
- Page 4, Table 2, Item "Access to healthcare": The text "16 (21.3%)" appears under the "With insomnia (n=48)" column for women. 16/48 is 33.3%, not 21.3%. Please double-check this value and percentage calculation. Correction:Re-reading the question, it's "have you ever had to go without healthcare because you didn't have a way to get there?". This seems related to transportation difficulties. Let's check the women's data again: Women with insomnia: 16 reported this barrier (p=0.039 vs women without). 16/48 = 33.3%. Women without insomnia: 2 reported this barrier. 2/27 = 7.4%. The percentage presented in the table (21.3%) seems incorrect for 16 out of 48. Please verify this calculation.
- It will be helpful if the authors could add future directions in relation to the present findings after the conclusion section.
Author Response
Page 1, Abstract: "February 2022-September 2023" - Clarify if this is the data collection period for the parent study or the period during which this secondary analysis occurred. Data collection period seems more relevant.
- The following sentence has been amended in the abstract on page 1: The parent study enrolled adults stabilized on buprenorphine from February 2022–September 2023.
The authors are required to justify why did they particularly chose Buprenorphine in this study although they mentioned that both Buprenorphine and methadone are the first line treatment for OUD in the introduction section. It is also preferred to give a brief introduction about Buprenorphine usage and the nature and side effects of the drug.
- We appreciate this comment from the reviewer. The following has been added on page 1: Both methadone and buprenorphine are equally effective medications at achieving positive OUD treatment outcomes. In the United States, buprenorphine can be prescribed in an outpatient clinic setting, as opposed to methadone that can only be dispensed by outpatient treatment programs abiding by substantial federal regulations. Thus, the expansion of buprenorphine access and utilization remains a critical public health strategy to address the ongoing overdose crisis, given the significant reduction in overdose risk that OUD treatments with medications (i.e., methadone, buprenorphine) render, compared to OUD treatments without medication [4].
There are multiple inconsistencies in the reported sample sizes throughout the manuscript, which significantly hinders clarity and reproducibility.
- Abstract: States N=127 overall participants.
- Results Text (Page 4): States "Participants (N=130) were primarily women (n=78; 60.0%)..." Calculation: 78/130 = 60%. Okay.
- Table 1 Header: States Women N = 77, Men N = 52. Total = 77 + 52 = 129.
- Table 1 Data: Sum of women with/without insomnia = 49 + 28 = 77. Sum of men with/without insomnia = 27 + 25 = 52. Total = 129.
- Table 2 Header: States Women N = 75, Men N = 52. Total = 75 + 52 = 127.
- Table 2 Data (Women): Sum of women with/without insomnia = 48 + 27 = 75. Matches header.
- Abstract & Results Text (Insomnia numbers): Abstract states n=75 (59.0%) met criteria for insomnia. Results text (page 4) states n=75 (59.0%). However, adding the numbers from Table 1 (which aligns with the total N=129) gives 49 women + 27 men = 76 participants with insomnia.
- Action Required:The authors must carefully reconcile these numbers. Please verify the final analytic sample size (is it 127, 129, or 130?) and ensure all reported numbers (total N, N for women, N for men, N with/without insomnia, Ns in tables) are consistent throughout the Abstract, Methods, Results text, and Tables. Explain any participant exclusions that might lead to differing Ns between analyses or tables, if applicable. The discrepancy between 75 and 76 participants meeting insomnia criteria also needs correction and consistent reporting.
- We deeply apologize for the lack of consistency across numbers. We have corrected for this across the manuscript. The correct sample size is 130, and the number of women with insomnia vs. no insomnia is 50 and 28 respectively, and the number of men with insomnia vs no insomnia is 27 and 25 respectively. The numbers were correct for men, but numbers for women were adjusted and the statistics were re-run accordingly. We also updated the percentages and p-values; notably, the main findings remain the same with the updated analyses.
The authors appropriately state in the Methods (Section 2.3) that no adjustments for multiple comparisons were made due to the exploratory nature of the study. This limitation should be more strongly emphasized in the Discussion section. Acknowledge that the significant findings (p < .05) should be interpreted with caution due to the lack of correction and warrant replication in larger studies. Briefly mentioning the number of tests performed could also add context for the reader regarding the potential for chance findings.
- The following has been added as a limitation on page 6: Finally, findings are limited by the study’s small sample size and lack of corrections for multiple comparisons. Thus, statistically significant findings should be interpreted with caution.
Page 1, Introduction, Para 4, Sentence 3: "...main contributor to return to use and treatment discontinuation [6] [5]." The previous sentence references factors linked to treatment attempts [5]. It seems possible the order should be [5, 6] here as well. Please double-check that the content cited aligns precisely with the reference numbers used.
- We apologize for this citation error. This has been addressed on page 1.
Also in Page 5, Discussion, Para 3, Sentence 3: "...interplay of economic, social, and cultural factors [19] [20]."
- We apologize for this citation error. This has been addressed on page 5.
Page 4, Table 2, Item "Access to healthcare": The text "16 (21.3%)" appears under the "With insomnia (n=48)" column for women. 16/48 is 33.3%, not 21.3%. Please double-check this value and percentage calculation. Correction: Re-reading the question, it's "have you ever had to go without healthcare because you didn't have a way to get there?". This seems related to transportation difficulties. Let's check the women's data again: Women with insomnia: 16 reported this barrier (p=0.039 vs women without). 16/48 = 33.3%. Women without insomnia: 2 reported this barrier. 2/27 = 7.4%. The percentage presented in the table (21.3%) seems incorrect for 16 out of 48. Please verify this calculation.
- We apologize for this calculation error. This has been addressed in Table 2.
It will be helpful if the authors could add future directions in relation to the present findings after the conclusion section.
- The following has been added on page 6: Future work should continue to explore such factors and how these factors can be addressed in precision medicine interventions to meet the unique biopsychosocial needs of men and women with OUD.
Round 2
Reviewer 2 Report
Comments and Suggestions for Authors
No further comments are needed.